# Time series analysis of total and direct associations between high temperatures and preterm births in Detroit, Michigan

Carina J Gronlund [ID] ,[1] Alyssa J Yang,[2] Kathryn C Conlon,[3] Rachel S Bergmans [ID] ,[4] Hien Q Le,[5] Stuart A Batterman,[6] Robert L Wahl,[7] Lorraine Cameron,[8] Marie S O'Neill[9]

For numbered affiliations see end of article.

**Correspondence to**
Dr Carina J Gronlund;
gronlund@umich.edu

## ABSTRACT

**Objectives** Preterm births (PTBs) represent significant health risks, and several studies have found associations between high outdoor temperatures and PTB. We estimated both the total and natural direct effects (independent of particulate matter, ozone and nitrogen dioxide air pollutants) of the prior 2-day mean apparent temperature (AT) on PTB. We evaluated effect modification by maternal age, race, education, smoking status and prenatal care.

**Design and setting** We obtained birth records and meteorological data for the Detroit, Michigan, USA area, for the warm months (May to September), 1991 to 2001. We used a time series Poisson regression with splines of AT, wind speed, solar radiation and citywide average precipitation to estimate total effects. To accommodate multiple mediators and exposure-mediator interactions, AT inverse odds weights, predicted by meteorological and air pollutant covariates, were added in a subsequent model to estimate direct effects.

**Results** At 24.9°C relative to 18.6°C, 10.6% (95% CI: 3.8% to 17.2%) of PTBs were attributable to the total effects of AT, and 10.4% (95% CI: 2.2% to 17.5%) to direct effects. Relative excess risks of interaction indicated that the risk of PTB with increasing temperature above 18.6°C was significantly lower among black mothers and higher among mothers less than 19, older than 30, with late or no prenatal care and who smoked.

**Conclusion** This additional evidence of a direct association between high temperature and PTB may motivate public health interventions to reduce extreme heat exposures among pregnant women, particularly among those who may have enhanced vulnerability.

## INTRODUCTION

Preterm births (PTBs) are defined as births that occur before 37 gestational weeks. Babies born prematurely are at greater risk of infant death and impaired health, such as cerebral palsy and impairments of cognitive development, hearing, vision, respiration and digestion. In 2016, 10% of babies were born prematurely, with statewide rates ranging 7.8% to 13.7% in the USA.[1] Within-state disparities in preterm birth rates can be

greater: from 1990 to 2010, 18% of births in Detroit, Michigan, were PTBs.[2]

Certain risk factors are known to be associated with PTB, such as cigarette smoking, alcohol use, hypertension and diabetes.[1] Air pollution is associated with PTB,[3 4] and higher ambient temperature has been proposed as a risk factor for PTB.[5 6]

High ambient temperature could pose a risk for PTB through several biological pathways, including stress and dehydration pathways.[7] Several studies have evaluated possible associations between high ambient temperature and PTB, with mixed results.[5 6] Studies in cities in Australia, Quebec, China, Belgium, Italy, Spain and across the USA have all found significant positive associations between

hot temperatures within the week preceding delivery and PTB. [5 6 8–16] High ambient temperatures in the month of conception and the third trimester were also positively associated with PTB in Changsha, China.[17 18] However, earlier, rigorously conducted time series analyses of temperature and PTB in Germany and in London, England, did not find associations.[19 20] Likewise, in Chicago, Illinois, Porter *et al* found no effect of the July 1995 heat wave on gestational length.[21] Sources of heterogeneity among these effect estimates may include differences in study design, prevailing climate and regional adaptation, population structure, exposure assessment, critical windows of exposure considered and methods of gestational age assessment.[12]

**Strengths and limitations of this study**

► We expand on a growing literature demonstrating total effects of temperature on preterm birth (PTB) by examining mediation of the total temperature effect by air pollutants.
► We did not have geographic identifiers for the study participants at a level as fine as residential address or indoor exposure information, and we therefore assigned time-varying environmental exposures at an outdoor, citywide level.
► A low prevalence of residential air conditioning (32%) in the study time period suggests that our exposure assignment using outdoor exposures alone may have more accurately reflected total daily exposures than in more recent studies in other geographies.
► We account for multiple mediators and potential exposure-mediator interactions by applying a straightforward technique — the use of inverse odds weights — to separate the total effects of temperature on PTB from the direct effects.
► We used methods examining potential synergistic effects, as opposed to just examining statistical interactions, by other characteristics that confer vulnerability to PTB.

Studies often control for air pollution exposure, although the argument has been made that treating air pollutants as confounders of temperature-health associations is inappropriate given that high temperatures can contribute to increased concentrations of some air pollutants, and both temperature and air pollution are affected by sunlight.[22] Differences in the manner in which air pollutants are accounted for in the temperature-PTB modelling may also account for some of the heterogeneity observed in this literature.

We performed a time series analysis to investigate the association between high apparent temperature (AT) and PTB in the Detroit, Michigan area, estimating both the total effects of temperature as well as the natural direct effects. To estimate natural direct effects, we excluded potential mediation effects by the air pollutants ozone, particulate matter with an aerodynamic diameter of less than 10 micrometres ($PM_{10}$), and nitrogen dioxide ($NO_2$) using an inverse odds weighting technique. We further examined whether the maternal risk factors of age, race, education, smoking and level of prenatal care modified the association between high AT and PTB.

## METHODS
### Outcome variable
Birth outcome data were derived from an electronic database of birth certificate records requested from the Michigan Department of Health and Human Services Division for Vital Records and Health Statistics and subsequently limited to live, singleton births that occurred from 1 January 1991 to 31 December 2001 in ZIP codes within 4 km of one of three air quality monitors (Allen Park, Linwood and East 7 Mile) in and bordering Detroit, Michigan.[4] In the final data set, births were categorised as term versus PTB, defined as births that were less than 37 gestational weeks. We limited the data set to births that occurred May to September in order to focus on heat exposures, for a total of 9053 births.

### Identification of confounders and mediators
A directed acyclic graph (DAG) was constructed to define the causal framework and elucidate potential mediating effects of air pollutants in the association between

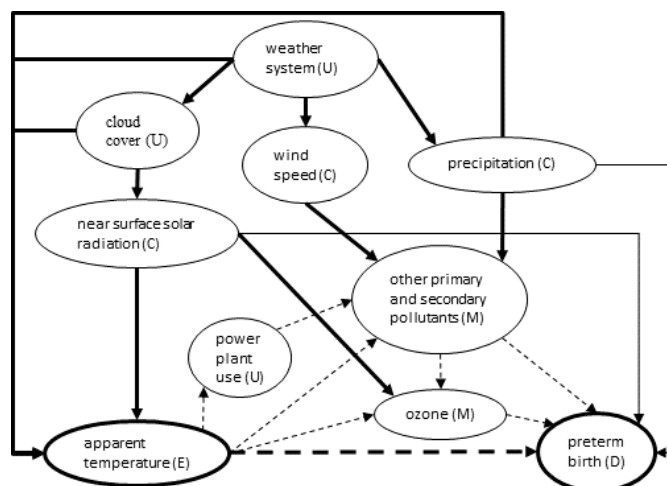

**Figure 1** Directed acyclic graph of the causal framework for mediation of the association between apparent temperature and preterm birth by air pollutants. U = unmeasured, C=confounder, M=mediator, E=exposure of interest, D=outcome. Thick solid lines known associations. Thin solid lines indicate suspected associations. The thick dashed line indicates the direct effect pathway being tested. The thin dashed lines indicate the indirect effect pathways being tested. Other primary and secondary pollutants include particulate matter less than 10 microns in aerodynamic diameter and nitrogen dioxide.

AT and PTB (figure 1). In this DAG, AT is the exposure of interest and PTB the outcome of interest. Given suspected associations between PTB and precipitation[23] and between PTB and vitamin D, for which one source is solar radiation,[24] as well as causal meteorological relationships between AT, solar radiation and precipitation via a parent weather system, precipitation and solar radiation are potential confounders of the AT-PTB association. In this DAG, primary and secondary air pollutants are the hypothesised mediators of the temperature-PTB association given that: (1) they have been associated with the health outcome in other locations[3] and in this data set in the third trimester,[4] (2) high temperature enhances the rate of ground-level ozone formation and (3) power plant emissions increase when temperatures rise to accommodate additional demand for air conditioning.[25] Furthermore, the weather system is a causal parent to both AT and the pollutants and therefore a potential confounder given that meteorological conditions such as wind speed, precipitation and cloud cover can affect pollutant formation and exposure as well as sunlight, which increases AT and promotes ozone formation.[26] Therefore, a model characterising the *total effect* of AT would need to include the confounders of solar radiation, wind speed and precipitation but not include pollutants. A model characterising the *direct effect* of AT, independent of air pollution mediation, would need to include these mediators. The modelling used to characterise direct and indirect effects is further described in the Statistical Analysis section below.

## Exposure variables

Daily mean temperature, dew point temperature and wind speed were obtained from the National Center for Environmental Information Integrated Surface Daily Lite database of daily weather parameters from weather stations worldwide.[27] Data from the Detroit City Airport was used due to its proximity to the mothers' ZIP codes of residence. To better represent thermal discomfort, we used AT rather than air temperature. AT is similar to heat index, and increases with both air temperature and relative humidity. AT was calculated using the following formula: ($-2.653 + 0.994\times$(temperature in °C)$+0.0153\times$(dew point temperature in °C)$^2$).[28][29] From Detroit City Airport, 11% and 15% of temperatures and dew points were missing, respectively, so data from Detroit Metropolitan Airport were used to replace these missing data. Among non-missing values, daily AT from Detroit City Airport was highly correlated with that from Detroit Metropolitan Airport (Pearson's correlation coefficient=0.98).

The total amount of direct and diffuse solar radiation received on a horizontal surface during each 60 min period at the Detroit City Airport was retrieved from the National Solar Radiation Database.[30] These data were modelled from meteorological data including cloud cover, aerosol and ozone data from sources such as sun photometers, satellites and albedo data.[31] We further estimated daily means from the hourly solar radiation values.

Daily precipitation totals were obtained from Oregon State University's PRISM Climate Group.[32] These are modelled at a 4 km resolution based on observations and a climatologically-aided interpolation process. Rasters were cropped to the City of Detroit and daily citywide averages were calculated.

Daily 8 hour maximum ozone and daily mean $NO_2$ and $PM_{10}$ concentrations were obtained from the Environmental Protection Agency for all Wayne County, Michigan monitors and averaged by day and pollutant.[33] Only daily monitor values for which at least a single daily or 18 of the 24-hourly values were available were retained, resulting in substantial missingness, which was addressed in the statistical analysis below.

Birth certificate data also included date of birth and maternal age group (16 to 19, 20 to 29 and over 30), race (black or white), smoking status (smoker vs non-smoker), education level (less than high school vs high school or higher) and level of prenatal care (prenatal care vs late or no prenatal care), which were used in analyses of effect modification.

## Patient and public involvement

Patients and the public were not involved in the design or planning of the study.

## Statistical analysis

Case-crossover analysis is commonly used in studies of PTB and temperature.[12][13][34] Because PTB is not a rare event in this particular population, the case-crossover ORs would not approximate risk ratios. Therefore, we used a

time series design with Poisson regression, controlling for seasonal and long-term variations in PTB counts with a cubic b-spline for day-of-year, with 5, two or eight knots, and a cubic b-spline for year with two knots. AT exposure was characterised as 2-day mean AT, or the mean of the AT values on the day of and the day before birth. For AT and the covariates, non-linearity was considered by initially modelling each as a b-spline with three knots and selecting a single knot for subsequent modelling of the covariate as a piecewise linear spline when substantial non-linearity was visually evident. The 95% CIs were constructed from the 2.5th and 97.5th percentiles of 500 bootstrapped samples. In sensitivity analyses, given its wide usage for this research question, a time-stratified case-crossover design was used with time strata defined as 2- or 3-week periods.

To account for missing air pollutant values, we conducted multiple imputations using chained equations. We used a more general model including lag days 0 to 2 of the above meteorological and pollutant values. The air pollutant values were well-predicted by lag days 0 to 2 meteorology and air pollution values, and from examinations of trace plots, or scatter plots of successive parameter estimates, we determined that a total of three imputations following two burn-ins was sufficient. All subsequent analyses were conducted on each of the three resulting data sets.

To estimate the total effects of AT, we included terms for solar radiation, wind speed and precipitation in the Poisson model, which blocked all of the paths in the DAG from AT to PTB that did not pass through the mediators (figure 1). However, to estimate natural direct effects, we used a more generalisable technique — inverse odds weighting — given that we had multiple mediators and as well as potential exposure-mediator interactions.[35] In this technique, we first fit a standard linear regression of 2 day mean AT (meanAT01) on the air pollutant mediators and covariates (solar radiation, wind speed, precipitation). We then estimated:

$$inverse\ odds = 1/exp(predicted\ meanAT01/\sigma^2) \quad \text{Equation 1}$$

where $\sigma^2$ was the model mean squared-error. The inverse odds were then used as weights in the subsequent analysis of PTB and AT with the other covariates but not the air pollutants. The weights render AT independent of the mediators, thereby allowing the estimation of AT separately from the effects of the air pollutant mediators on PTB. Total and direct effects were calculated from each of the three imputed data sets and then each averaged.

To assess the public health impact of AT on PTB, we estimated the attributable fraction (AF), or the percent of PTB attributable to high AT among women exposed to the high AT. The AF was estimated as: $1/(1-RR)*100\%$ where the RR (relative risk) was for the 95th percentile (24.9°C) relative to the 50th percentile (18.6°C) of May to September 2-day mean AT. Indirect effects were estimated as the difference between the total effects and the

direct effects. The 95% CIs were constructed from the 2.5th and 97.5th percentiles of 500 bootstrapped samples.

To understand how the effects varied among maternal subgroups, or effect modification of the PTB-AT association, we simultaneously included interaction terms between the AT spline and the indicator variables for black race, age 16 to 19, age 30 years or older, low prenatal care (late or no prenatal care), current smoker and no high school education. We expanded the data set such that we had rows for each unique combination of date and daily exposures, race, age group, education, prenatal care and smoking status. Given the large number of zero counts, we specified a negative binomial distribution rather than a Poisson distribution. For public health significance, we were interested in the absolute rather than the relative increase in PTB risk due to synergistic effects between temperature and each modifier of interest. Therefore we focused on *additive*, rather than *multiplicative*, interactions. Relative excess risk due to interaction (RERI)[36] was calculated for each potential modifier as:

$$RR_{AT=95th,EM=1} - RR_{AT=95th,EM=0} - RR_{AT=50th,EM=1} + 1 \quad \text{Equation 2}$$

where EM was the effect modifier of interest and the denominator of each RR was the risk at the 50th percentile of AT (18.6°C) and the absence of the modifier (EM=0). RERI CIs were bootstrapped from 1000 samples. Analyses were conducted in SAS 9.4 (Copyright 2016, SAS Institute Inc, Cary, North Carolina, USA) using PROC GLIMMIX, which allows for multiple splines in a Poisson regression. Figure 2The exposure-response graph was constructed in R (R Foundation for Statistical Computing, Vienna, Austria) using the dlnm package[37] following modelling using the glm.nb function in the MASS package.[38]

## RESULTS

There were 9053 singleton PTBs in this Detroit-area sample, May to September, 1991 to 2001 (table 1). There were fewer PTBs in September compared with the other months, and consistent with Detroit's population decline,

**Table 1** Demographics of preterm births in Detroit, Michigan study area, May to September, 1991 to 2001

|  | N | % |
|---|---|---|
| Total | 9053 | 100 |
| Month | | |
| May | 1918 | 21.2 |
| June | 1944 | 21.5 |
| July | 1903 | 21.0 |
| August | 1754 | 19.4 |
| September | 1534 | 16.9 |
| Year (two equal time periods) | | |
| 1991–1995 | 4708 | 56.2 |
| 1996–2000 | 3670 | 43.8 |
| Maternal race | | |
| White | 2443 | 27.0 |
| Black | 6433 | 71.1 |
| Other | 177 | 1.9 |
| Maternal smoking status | | |
| Non-smoker | 6621 | 73.1 |
| Smoker | 2345 | 25.9 |
| Missing | 87 | 1.0 |
| Level of prenatal care | | |
| Prenatal care | 5037 | 55.6 |
| Late or no prenatal care | 2960 | 32.7 |
| Missing | 1056 | 11.7 |
| Maternal age | | |
| 16–19 years | 1782 | 19.7 |
| 20–29 years | 4821 | 53.2 |
| ≥30 years | 2450 | 27.1 |
| Maternal education | | |
| Less than high school | 3485 | 38.5 |
| High school or higher | 5451 | 60.2 |
| Missing | 117 | 1.3 |

the number of PTBs declined with time. Considering individual characteristics, 27.0% of the PTBs were among white mothers and 71.1% were among black mothers. A majority of the mothers were non-smokers (73.1%), had prenatal care (55.6%), were 20 to 29 years of age (53.2%) and had at least a high school education (60.2%) (table 1).

On average, AT exposure was 18.0°C, and among the days when AT was greater than the time-period-specific daily median of 18.6°C (ie, for the upper end of the temperature spline), AT averaged 1.5°C higher. A total of 53% of the cases occurred on days with no rain, so the geometric mean precipitation was 0.4 mm. The mean ozone concentration was 44.8 ppb and the maximum was 102 ppb (table 2). In examining the daily time series of ozone over the study period, the 3-year running averages of the fourth highest daily 8 hour maximum value

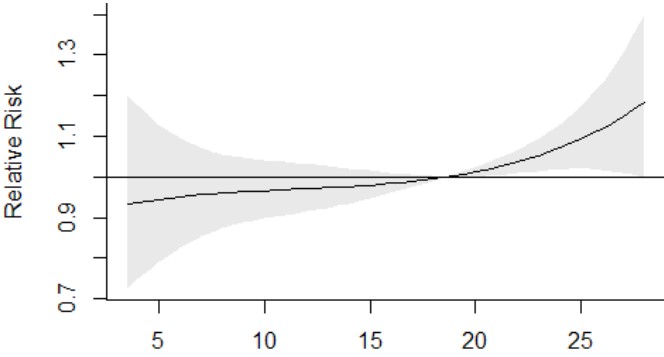

**Figure 2** Association between mean apparent temperature (AT) over lag days 0 to 1 and preterm birth, modelling AT as a b-spline with 3 df.

**Table 2** Daily exposures among the PTB cases, Detroit, Michigan area, May to September, 1991 to 2001

|  | N | Median | Mean | Min | Max |
|---|---|---|---|---|---|
| Two-day mean apparent temperature, °C | 9053 | 18.6 | 18.0 | 3.1 | 28.0 |
| Two-day mean apparent temperature 18.6°C and above | 9053 | 0.1 | 1.5 | 0.0 | 9.4 |
| Mean solar radiation (W/m²) | 9053 | 226.4 | 217.7 | 38.1 | 352.8 |
| Total precipitation (mm)* | 9053 | 0.0 | 0.4 | 0.0 | 56.7 |
| Mean wind speed (m/s) | 9053 | 3.6 | 3.6 | 0.1 | 8.0 |
| Maximum 8 hour average ozone (ppb) | 9039 | 43.0 | 44.8 | 3.2 | 102.0 |
| Mean particulate matter, 10 μm or less (μg/m³)* | 8331 | 39.0 | 38.5 | 8.0 | 158.0 |
| Mean nitrogen dioxide (ppb)* | 8956 | 18.4 | 19.7 | 0.0 | 72.8 |

*Geometric means are provisded. Values were natural-log-transformed in the regression analyses.
PTB, preterm birth.

of ozone ranged from 77 to 92 ppb, suggesting that the (current) National Ambient Air Quality Standards (70 ppb) had been exceeded in each year of the study.

In a crude model of the association between PTB and AT where AT was modelled flexibly as a b-spline with 3 df, we found a non-linear association, with approximately null effects below 18°C and an increasingly stronger positive association at higher temperatures (figure 2). To address potential sensitivity of the results to control for season, we varied the df in the day-of-year term, using 2, 5 and 8 df. The results were not highly sensitive to this choice, with the point estimates of the percentages of PTB attributable to AT ranging only from 8.1% (8 df) to 9.9% (2 df) among pregnant women exposed to days when AT was 24.9°C versus exposure on 18.6°C days (table 3). In case-crossover analyses, the resulting ORs were also statistically significantly greater than 1.0, regardless of whether the time strata were 2-week, 3-week or 1-month periods (online supplementary table 1). However, the ORs were greater in magnitude than the risk ratios estimated in the time series design, which is expected, given that ORs overestimate risk ratios when events are not rare.

The total effects were estimated as 10.6% (95% CI: 3.8% to 17.2%) of PTBs attributable to AT among pregnant women exposed to a day when AT was 24.9°C versus

exposure on an 18.6°C day. Accounting for mediation of this effect by $PM_{10}$, ozone and $NO_2$ by using inverse odds weights, the direct effect of AT on preterm birth was decreased to 10.4% (95% CI: 2.2% to 17.5%), although the difference between the two values was not statistically significant (indirect effect=0.3%, 95% CI: −1.7% to 2.6%).

In examining the RERI from interactions between AT and maternal characteristics, black race was found to be protective, with an RERI significantly less than 0 (table 4). The magnitudes of RERIs are not meaningful, but this result indicates that the association between high temperature and PTB is weaker among women of black race than non-black race. In contrast, RERIs for age 16 to 19 years, age >30 years, low prenatal care and tobacco smoking were all significantly greater than 0, indicating that the association between PTB and high temperature is stronger among women with these characteristics.

## DISCUSSION

The strong association between short-term temperature exposure and PTB in Detroit, Michigan, USA was consistent with several other studies' observed associations between PTB or diminished gestational age and temperature exposures within the week prior to delivery at regionally high

**Table 3** Relative risk of preterm birth and per cent of preterm births attributable to 2-day mean apparent temperature (AT) on a 24.9°C day versus an 18.6°C day, Detroit, Michigan area, May to September, 1991 to 2001

| Model | Knots in day-of-year spline | Covariates | Relative risk (95% CI) | Per cent attributable (95% CI) |
|---|---|---|---|---|
| 1 | 2 | None | 1.11 (1.02 to 1.09) | 9.9 (2.4 to 16.4) |
| 2 | 5 | None | 1.11 (1.03 to 1.20) | 9.8 (2.6 to 16.5) |
| 3 | 8 | None | 1.09 (1.01 to 1.18) | 8.1 (1.0 to 17.6) |
| 4 | 5 | Solar radiation, wind speed, precipitation | 1.12 (1.04 to 1.21) | 10.6 (3.8 to 17.2) |
| 5 | 5 | Solar radiation, wind speed, precipitation, inverse-odds weights* | 1.12 (1.02 to 1.21) | 10.4 (2.2 to 17.5) |

*Inverse-odds weights calculated from the predicted odds of AT given lag day 0 and 1 of ozone, $PM_{10}$ and $NO_2$.
$NO_2$, nitrogen dioxide; $PM_{10}$, 10 micrometres.

**Table 4** Relative excess risk due to interaction (RERI) for interactions of 2-day mean apparent temperature on a 24.9°C day versus an 18.6°C day with maternal characteristics, Detroit, Michigan area, May to September, 1991 to 2001

| Characteristic | RERI | 95% CI |
| --- | --- | --- |
| Black race | −1.5 | -1.9 to 1.0 |
| Age 16–19 years | 0.50 | 0.22 to 0.76 |
| Age >30 years | 0.34 | 0.08 to 0.65 |
| Low prenatal care | 0.44 | 0.19 to 0.65 |
| No high school | 0.19 | −0.06 to 0.43 |
| Smoker | 0.52 | 0.29 to 0.76 |

With five knots in the day-of-year spline.

temperatures. Regions in which these associations have been found include Central Australia (8.3% at 40°C daily maximum temperature),[11] Northern California, USA (11.6% increase for a 5.6°C increase in weekly average AT in the warm season),[34] Alabama, USA (32.4% increase with two consecutive days of daily mean temperatures above the 98th percentile),[39] Barcelona, Spain (5-day reduction in average gestational age with heat index above the 99th percentile),[8] Rome, Italy (1.9% increase per 1°C increase in maximum AT in the prior 2 days and 19% increase during heat waves),[40] Brisbane, Australia (13% to 100% increase during heat waves), in an aggregated sample of 12 US cities (12% to 16% increase with 2.8°C increase in prior week),[12] in Southern China (7% increase with previous-week temperatures above 95th percentile)[10] and in a multi-city USA sample (2% increased PTB risk with extreme heat in the prior 4 days).[41]

The precise biological mechanisms by which high ambient temperature might increase risk for PTB are unclear, although psychosocial stress[7] and dehydration pathways are plausible.[7] Stress increases the levels of cortisol and epinephrine, potentially leading to the secretion of placental corticotropin-releasing hormones (CRH). Placental CRH can then activate the fetal hypothalamic-pituitary-adrenal axis, which could prompt the fetal expression of cortisol and dehydroepiandrosterone-sulfate and placental release of estriol and prostaglandins, potentially triggering PTB.[42] Dehydration due to high temperatures and sweating could also reduce blood flow to the uterus and induce a greater release of antidiuretic hormone and oxytocin, which could trigger labour onset.[43]

One strength of our study may be a tighter correspondence between the true ambient temperatures experienced by our sample and the measured outdoor ambient AT given the low air conditioning prevalence in the Detroit area during the time period. Specifically, in 1993, the last year for which county-specific American Housing Survey data were available in the Detroit area, only 32% of Wayne County households had central air conditioning.[44] Furthermore, central air conditioning prevalence was low in this

region despite a 'hot-summer humid continental climate' Köppen climate classification,[45] allowing a fairly wide range of warm season AT exposure over which AT-PTB associations could be examined. In contrast, regions where associations between temperature and PTB were null in rigorously conducted daily time series studies included Brandenburg and Saxony, Germany,[20] and London, UK,[19] which are in 'temperate oceanic climates,' where all months have average temperatures below 22°C.[45] Additionally, Guo et al[10] found an association between previous-week temperature and PTB only in the 'hot' region of China, defined as having annual average temperatures 19°C or higher. Furthermore, in a survival analysis of previous-week temperature and PTB in 18 European cities, the pooled effect estimates were null, and no individual city results were presented.[46] Again, with the exception of four cities, these European cities were all in cooler climates where all months have average temperatures below 22°C.[45] This suggests that in regions where the threat of extremely high temperatures is rare, PTB is not triggered at warm temperatures, regardless of relative temperature thresholds, even at the same absolute temperatures at which heat-associated PTB is experienced in warmer climates. This could be due to differences in emotional stress[47] or physiological stress responses to warm temperatures between climates. Alternatively, the heterogeneity in effect estimates could be due to regional differences in the misclassification of the true individually-experienced temperatures by outdoor temperatures. Finally, in the 18-city European study, the PTB rate was only 5%, suggesting that PTB aetiologies may differ between Europe and the USA, in which case the pregnancies in the European cohorts could have been less susceptible physiologically to high temperature.[46] Other strengths of our study include our consideration of near-surface solar radiation and weather conditions as confounders of the total effects of AT, air pollutants as mediators rather than confounders and estimation of the direct effects of AT as distinct from those mediated by air pollutants. By using an analysis technique that was 'agnostic' to exposure-mediator interactions and could accommodate multiple mediators,[35] we found that most if not all of the effect of AT on PTB is direct, and not mediated by daily changes in air pollution concentrations. However, this mediation analysis technique tends to overestimate the CIs around the indirect effects,[35] which may account for our finding a null indirect effect of air pollutants on PTB when previous research has in fact identified significant associations between short-term increases in air pollution and PTB.[48] We may have also underestimated the air pollutant effects because we were only examining the indirect effects of AT through an air pollutant pathway rather than the total effects of air pollution and because we only considered $PM_{10}$ rather than $PM_{2.5}$.

Another strength of our study is that we reported RERIs, rather than merely reporting statistical interactions in the models. In doing so, we provided evidence for synergistic effects of high temperature with the independent risk factors for PTB of prenatal care, smoking status and age. Interestingly, in this majority-black population, the risk of

PTB with high temperatures was actually lower among black mothers, after controlling for age, education, smoking status and prenatal care.

Limitations include the fact that we were not able to distinguish between spontaneous and medically-indicated PTBs. Considering that medically-indicated preterm deliveries are unlikely to be related to temperature, this limitation affects the generalisability of our relative risks to populations where the relative percentages of spontaneous versus medically-indicated PTB differ. We also did not have information on how much earlier the birth was, for example, a 1 day premature birth versus a 6 week premature birth. This prevented us from including an offset for the population of pregnancies at-risk of PTB.[49] Our model included a spline for year and spline for day-of-year to attempt to capture within-season variations in PTBs. Varying the df in the day-of-year term from 2 to 8 had only a mild effect on the relative risks, although this would not have captured sub-weekly changes in PTBs. Another limitation is our dependence on last menstrual period rather than ultrasound-derived gestational age. Last menstrual period tends to result in more births being classified as preterm, particularly among African-Americans.[50] This limitation, assuming it was not correlated with temperature, would bias our results towards the null.

An additional limitation is that the prevalence of characteristics enhancing vulnerability to PTB may have changed since the study period, thereby decreasing the generalisability of these results to the present-day population of this region. Future research should employ present-day cohorts of pregnant women, for which a denominator of total pregnancies is therefore available, linked with refined temperature exposure measurements, including indoor and neighbourhood temperature exposure estimates. These research refinements will better characterise the thermal exposures and severity of heat-induced PTB, or more specifically, gestational length, and better identify which pregnancies are particularly vulnerable to early parturition on hot days.

Despite the aforementioned limitations, given the evidence from this and other studies, pregnant women, in addition to older adults, should be considered as a group vulnerable to short-term heat health effects when considering housing and climate adaptation measures in Detroit and similar or warmer climates.

**Author affiliations**
$^1$Survey Research Center, University of Michigan Institute for Social Research, Ann Arbor, Michigan, USA
$^2$Urban Indian Health Institute, Seattle, Washington, USA
$^3$Public Health Sciences, University of California Davis, Davis, California, USA
$^4$Psychiatry, University of Michigan Medical School, Ann Arbor, Michigan, USA
$^5$Toxicology and Risk Assessment, Chemours Co, Wilmington, Delaware, USA
$^6$Environmental Health Sciences, University of Michigan, Ann Arbor, Michigan, USA
$^7$Surveillance and Program Evaluation Section, Michigan Department of Health and Human Services, Lansing, MI, USA
$^8$Michigan Climate and Health Adaptation Program, Michigan Department of Health and Human Services, Lansing, Michigan, USA
$^9$Environmental Health Sciences and Epidemiology, University of Michigan, Ann Arbor, Michigan, USA

**Contributors** MSO, SAB, HQL, RLW and LC obtained the data or resources, critiqued the analysis and reviewed the manuscript. RSB and KCC assisted with preliminary analyses, critiqued the analysis and reviewed the manuscript. AJY performed and drafted preliminary analyses and reviewed the manuscript. CJG directed preliminary analyses, revised the analysis and revised the manuscript. We thank Leah Comment for early stage data management and analysis assistance. We also thank Patricia Maina for her contribution to the preterm birth and heat literature review and Sung Kyun Park for early stage advice.

**Funding** This work was supported by a Michigan Institute for Clinical and Health Research Postdoctoral Translational Scholars Fellowship (2UL1TR000433-06), National Institute of Environmental Health Sciences grants K99ES026198 and P30ES017885, Cooperative Agreement Number EH001124 from the Centers for Disease Control and Prevention (CDC), National Science Foundation grant 1520803, and CDC/National Institute for Occupational Safety and Health grant T42 OH008455. Its contents are solely the responsibility of the authors and do not necessarily represent the official views of the CDC or the Michigan Department of Health and Human Services. None of the funders participated in the design, collection, analysis or interpretation of the data.

**Competing interests** None declared.

**Patient consent for publication** Not required.

**Ethics approval** The Michigan Department of Health and Human Services Institutional Review Board (IRB) (study number 201302–03-XA) and the University of Michigan Institutional Review Board (study number HUM00071694) determined the study exempt from IRB review per Title 45 Code of Federal Regulations 46.101. (b) – research involving collection of existing data and information is recorded by the investigator in such a manner that subjects cannot be identified.

**Provenance and peer review** Not commissioned; externally peer reviewed.

**Data availability statement** Birth outcome data were derived from birth certificate records kept by the Michigan Department of Health and Human Services (MDHHS). These confidential data may be obtained from the MDHHS Division for Vital Records and Health Statistics following completion of a data use agreement and IRB approval. The exposure data are publicly available and were obtained as described in the Methods. The processed exposure data may be requested from C. Gronlund ( gronlund@umich.edu).

**ORCID iDs**
Carina J Gronlund http://orcid.org/0000-0002-0533-745X
Rachel S Bergmans http://orcid.org/0000-0001-5740-6691

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
