## [Reviewer comments · BMJ Open]

ARTICLE DETAILS

TITLE (PROVISIONAL)	A time series analysis of total and direct associations between high temperatures and preterm births in Detroit, Michigan
AUTHORS	Gronlund, Carina; Yang, Alyssa; Conlon, Kathryn; Bergmans, Rachel S.; Le, Hien; Batterman, Stuart; Wahl, Robert; Cameron, Lorraine; O'Neill, Marie

VERSION 1 – REVIEW

REVIEWER	Qihong Deng Central South University, Changsha, China
REVIEW RETURNED	08-Jul-2019

GENERAL COMMENTS	A good work about this topic. I have the following suggestions for revising: TITLE. I suggest the authors to delete "total", because the direct association is more attractive. In fact, the manuscript indicated the association between temperature and preterm birth was basically not affected by the air pollution. In addition, please revise "temperatures" into "temperature" . INTRODUCTION 1. The 3rd paragraph "High ambient temperature could pose a risk for PTB through several biological pathways ...". I suggest the authors to remove this paragraph into DISCUSSION as "biological mechanism". Here, one or two sentences is OK. 2. About "Several studies have evaluated possible associations between high ambient temperature and PTB". There are several recent "long-term" studies also showed an association between high/low ambient temperature and PTB: (1) Zheng et al. An epidemiological assessment of the effect of ambient temperature on the incidence of preterm births: Identifying windows of susceptibility during pregnancy. Journal of Thermal Biology,2018,74:201-207. (2) Zhong et al. Preterm birth and ambient temperature: strong association during night-time and warm seasons. Journal of Thermal Biology,2018,78:381-390. I think the long-term association provides a good background or evidence to the short-term study. 3. The two paragraphs "Several studies have evaluated possible associations between high ambient temperature and PTB ..." and "However, earlier, rigorously conducted time series analyses of temperature and PTB ..." should be combined into one paragraph.
--

	4. I suggest these sentences "Additionally, studies often control for air pollution exposure, although the argument has been made that treating air pollutants as confounders of temperature-health associations is inappropriate given that ..." should be expanded into a separate paragraph to explain the necessary of "direct association" analysis. METHODS 1. "A directed acyclic graph (DAG) was constructed to define the causal framework and elucidate potential mediating effects of air pollutants in the association between AT and PTB (Figure 1)". I don't think this figure is good. I suggest the authors to delete it. RESULTS The results showed that the total association was 10.6% (95% CI: 3.8-17.2%) and the direct association was 10.4% (95% CI: 2.2-17.5%), which indicated that the association between temperature and PTB was not affected by the air pollution. This is good, but the authors should discuss this in DISCUSS, because some studies found that the air pollution was associated with PTB.
--	---

REVIEWER	Ana M. Vicedo-Cabrera London School of Hygiene and Tropical Medicine, London. UK.
REVIEW RETURNED	04-Aug-2019

GENERAL COMMENTS	Thanks for giving me the opportunity to review this interesting manuscript. The authors assessed the short-term association between temperature and preterm birth in Detroit. Although the research topic is not new, the authors incorporated the novelty of disentangling the direct from the total effects using DAGs. The study can be considered relevant to clarify the role of ambient temperature as trigger of preterm birth, in particular, to assess to which extent the observed association is due to a mediating effect of air pollutants. The manuscript is well written and well structured, and the references are appropriate. Given that I don't consider myself an expert on DAG, I will focus my review on the time-series modelling approach and the discussion of the results. I agree with the authors that time series analysis is an appropriate study design to address their research question. Overall, I consider that the analysis that the authors performed is correct. However, I wonder why the authors did not consider including a denominator on the pregnancies-at-risk (see in Darrow et al Epidemiology 20, 689–698. 2009, Schifano et al Env Int 2014, Vicedo-Cabrera et al Env Res 2014). It has been shown that the daily pool of individuals at risk (ongoing pregnancies at risk of being preterm) changes from one day to the other. By don't taking this into account, one might under or overestimate the association. I would suggest the authors to consider running a sensitivity analysis incorporating as denominator the number of daily pregnancies at risk (or even consider the "corrected" denominator developed in Vicedo-Cabrera 2015). The authors considered lag 01 of AT as main exposure term. It has been shown that the association can last up to 2-5 days after the exposure. I would suggest the authors to explore additional lagged terms as sensitivity analysis. Minor comments:
---

	 - Why the authors considered apparent temperature as temperature index, and not mean temperature or maximum apparent temperature, for example? This should be clarified in the text. - Figure 1 is a bit confusing; I would suggest changing the layout, possibly including colours. - It would be interesting to show the exposure-response curve representing the association. - The study period used is relatively old. One can expect that the observed impacts might have increased due to the increasingly warmer temperatures due to climate change. I would suggest the authors to discuss this point as an additional limitation. - Include a new reference for your interest: Sun et al. Ambient temperature and preterm birth: A retrospective study of 32 million US singleton births. Env Int 2019.
--	--

REVIEWER	Jennifer Zeitlin Inserm, Paris
REVIEW RETURNED	24-Aug-2019

GENERAL COMMENTS	Total and direct associations between high temperatures and preterm births in Detroit, Michigan This study has many strengths, starting with its research question, which is to assess whether higher temperatures affect risks of preterm birth. Given climate change and the importance of preterm birth as a public health issue, this question merits thorough investigation. The authors add to the previous literature by exploring whether air pollution is a mediator of the relationship between air temperature and preterm birth risk. Their data spanned 10 years between 1991 and 2001 in the Detroit area. This is a period when air conditioning was not as widely spread and this is also an interesting feature of this study. The manuscript is well written and the authors do a good job of putting their results into context using the scientific literature on temperature and preterm birth risk in the discussion. I am not a specialist in environmental epidemiology or the use of time series methods, so I cannot provide a review of these methods. As someone who is not a specialist in the methods used in the paper, I would have appreciated more time given to presenting the models, with further explanation of the methodological choices adopted (one ex. "generating three imputations following two burn-ins." - Why 3 and 2?). In particular, more explanation about the analyses of seasonal effects would be helpful. I did not understand whether the approach takes into consideration the seasonality of preterm birth or the seasonality of births more generally, or both. I would have liked more explanation of the test for this: (i.e. varying the df in the day-of-year term, using 2, 5, and 8 df). I also wondered how the highest temperatures were distributed across the months included in the study. I would also suggest that as the case-crossover design is often used for these analyses and the authors did carry out this analysis, their results should be included in a supplementary table. Also, having these estimates available would be good as a point of comparison with previous studies. Given that air pollution was not a mediator for the impact of air temperature, did the authors find that air pollution had an impact on PTB risk in this population independently of AT? Unfortunately the authors are not able to separate medically indicated from spontaneous preterm deliveries (probably about 20-30% of the total) which are less likely to be impacted by temperature
--

	(given the hypothesized mechanisms); This is mentioned in the limitations, but could the authors go further and indicate how they believe this limitation affects their estimation of the effect size and attributable risk? Finally, given the error of gestational age estimation using LMP in US birth certificate data, shown to be larger among Black women, how do the authors assess that misclassification error might affect their results? Minor comments Would it be possible to have the distribution by month in Table 1 and some information on trends over the 10 year period in PTB and temperature? Do the authors mean “all” instead of “any. “which blocked any of the paths in the DAG from AT to PTB that did not pass through the mediators (Figure 1)” and is this sentence needed? –it seem to be referring to consideration of classical confounding? One suggestion might be to indicate the variables in the DAG which the authors consider in their model and those which they don't. Note: I did not see ethics approval mentioned in this document. The data were publically available (as stated in the data sharing agreement section), but does the data sharing agreement require IRB approval. Please clarify given patient location identifiers.
--	---

REVIEWER	Duc Anh Ngo Nashville Metro Health, US
REVIEW RETURNED	20-Sep-2019

GENERAL COMMENTS	Total and direct associations between high temperatures and preterm births in Detroit, Michigan The paper examines the effects of the prior two-day mean apparent temperature (AT) on PTB and interaction between AT and maternal characteristics on the risk of PTB. The paper was written in a way that is difficult to follow and provide specific comments. There seems to be a lot of major methodological issues. Below are some examples. 1/ The major weakness is the lack of measurement of individual exposure to high temperature, and control for other factors that affect indoor temperature to which the women were directly exposed, even the outdoor temperature was the same. 2/ Causal framework (Figure 1) is too complex and difficult to interpret. It is unclear how this framework was operationalized in the analysis. For example, what are potential effect modifiers, moderators, or mediators for the association between AT and PTB. 3/ Method section mentioned a number of Confounders and Mediators (page 6), but what variables that were actually treated as Confounders and Mediators in the statistical analysis were not described. In Table 3, only covariates were listed, without defining what variables were confounders and what variables were mediators. No potential effect modifiers were described in the methods. 4/ Exposure definition: ‘prior two-day’ mean apparent temperature (AT) was mentioned in the Abstract, but not in the method. Furthermore, when the temperature was measured and why it would be related to pregnancy and the occurrence of PTB was not stated. Was the temperature measured when the women was pregnant? If so, during what period of pregnancy? 5/ It seems that the paper tested additive (as opposed to multiplicative) interaction between AT and maternal characteristics on the risk of PTB. This should be clearly described in the method.
---

VERSION 1 – AUTHOR RESPONSE

Reviewer #1:

TITLE

I suggest the authors to delete "total", because the direct association is more attractive. In fact, the manuscript indicated the association between temperature and preterm birth was basically not affected by the air pollution. In addition, please revise "temperatures" into "temperature".

We have revised the term 'temperatures' to 'temperature. We respectfully prefer not to follow the suggestion to delete the word "total" from the title. The terms "total" and "direct" have specific meanings in a mediation analysis, and even though we did not find significant indirect effects, we still feel that the title should indicate that we tested for them. We also feel that "total" and "direct" are terms that are accessible to a wide audience while still indicating that a mediation analysis was performed.

INTRODUCTION

1. The 3rd paragraph "High ambient temperature could pose a risk for PTB through several biological pathways ...". I suggest the authors to remove this paragraph into DISCUSSION as "biological mechanism". Here, one or two sentences is OK.

Thank you for this helpful suggestion. We have moved this third paragraph to the Discussion, slightly modifying the paragraph to read:

"The precise biological mechanisms by which high ambient temperature might increase risk for PTB are unclear, although psychosocial stress⁷ and dehydration pathways are plausible.. Stress increases the levels of cortisol and epinephrine, potentially leading to the secretion of placental corticotropin-releasing hormones (CRH). Placental CRH can then activate the fetal hypothalamic-pituitary-adrenal (HPA) axis, which could prompt the fetal expression of cortisol and dehydroepiandrosterone-sulfate (DHEA-S) and placental release of estriol and prostaglandins, potentially triggering PTB⁸. Dehydration due to high temperatures and sweating could also reduce blood flow to the uterus and induce a greater release of antidiuretic hormone and oxytocin, which could trigger labor onset⁹."

In the Introduction, we have made the following corrections:

"High ambient temperature could pose a risk for PTB through several biological pathways, including stress and dehydration pathways. Several studies have evaluated possible associations between high ambient temperature and PTB, with . . ."

2. About "Several studies have evaluated possible associations between high ambient temperature and PTB". There are several recent "long-term" studies also showed an association between between high/low ambient temperature and PTB:

(1) Zheng et al. An epidemiological assessment of the effect of ambient temperature on the incidence of preterm births: Identifying windows of susceptibility during pregnancy. *Journal of Thermal Biology*,2018,74:201-207.

(2) Zhong et al. Preterm birth and ambient temperature: strong association during night-time and warm seasons. *Journal of Thermal Biology*,2018,78:381-390.

I think the long-term association provides a good background or evidence to the short-term study.

Thank you so much. As you suggested, we have included these citations.

"High ambient temperatures in the month of conception and the third trimester were also positively associated with PTB in Changsha, China {Zheng, 2018;Zhong, 2018}."

3. The two paragraphs "Several studies have evaluated possible associations between high ambient temperature and PTB ..." and "However, earlier, rigorously conducted time series analyses of temperature and PTB ..." should be combined into one paragraph.

Thanks you! As suggested, we have combined these paragraphs.

4. I suggest these sentences "Additionally, studies often control for air pollution exposure, although the argument has been made that treating air pollutants as confounders of temperature-health associations is inappropriate given that ..." should be expanded into a separate paragraph to explain the necessary of "direct association" analysis.

As you suggested, we have added a paragraph break between these sentences.

METHODS

1. "A directed acyclic graph (DAG) was constructed to define the causal framework and elucidate potential mediating effects of air pollutants in the association between AT and PTB (Figure 1)". I don't think this figure is good. I suggest the authors to delete it.

We agree that this figure is confusing, and as reviewers 2, 3, and 4 suggested, we have modified it, changing the layout and including abbreviations for our hypothesized confounders and mediators as well as dashed lines of varying sizes to indicate the hypothesized direct and indirect pathways. In response to these reviewers, the following modifications were made to explain the DAG further.

"A directed acyclic graph (DAG) was constructed to define the causal framework and elucidate potential mediating effects of air pollutants in the association between AT and PTB (Figure 1). In this DAG, AT is the exposure of interest and PTB the outcome of interest. Given suspected associations between PTB and precipitation {Yu, 2018} and between PTB and vitamin D, for which one source is solar radiation {Dovnik, 2018 #3224}, as well as causal meteorological relationships between AT, solar radiation, and precipitation via a parent weather system, precipitation and solar radiation are potential confounders of the AT-PTB association. In this DAG, primary and secondary air pollutants are the hypothesized mediators of the temperature-PTB association given that: 1) they have been associated with the health outcome{Sun, 2015}, 2) high temperature enhances the rate of ground-level ozone formation, and 3) power plant emissions increase when temperatures rise to accommodate additional demand for air conditioning{Abel, 2017}. Furthermore, the weather system is a causal parent to both AT and the pollutants and therefore a potential confounder given that meteorological conditions such as wind speed, precipitation, and cloud cover affect pollutant formation and exposure as well as sunlight, which increases AT and promotes ozone formation{Environmental Protection Agency (EPA), 2018}. Therefore, a model characterizing the total effect of AT would need to include the confounders of solar radiation, wind speed, and precipitation but not include pollutants. A model characterizing the direct effect of AT, independent of air pollution mediation, would need to include these mediators. The modeling used to characterize direct and indirect effects is further described in the Statistical Analysis section below."

RESULTS

The results showed that the total association was 10.6% (95% CI: 3.8-17.2%) and the direct association was 10.4% (95% CI: 2.2-17.5%), which indicated that the association between temperature and PTB was not affected by the air pollution. This is good, but the authors should discuss this in DISCUSS, because some studies found that the air pollution was associated with PTB.

Thank you for this suggestion. In the Discussion, we have made the following modification:

"By using an analysis technique that was "agnostic" to exposure-mediator interactions and could accommodate multiple mediators{Nguyen, 2015}, we found that most if not all of the effect of AT on PTB is direct, and not mediated by daily changes in air pollution concentrations. However, this mediation analysis technique tends to over-estimate the confidence intervals around the indirect effects {Nguyen, 2015}, which may account for our finding a null indirect effect of air pollutants on PTB when previous research has in fact identified significant associations between short-term increases in air pollution and PTB{Stieb, 2019}. We may have also underestimated the air pollutant effects because we were only examining the indirect effects of AT through an air pollutant pathway rather than the total effects of air pollution and because we only considered PM₁₀ rather than PM_{2.5}."

Reviewer 2:

Thanks for giving me the opportunity to review this interesting manuscript. The authors assessed

the short-term association between temperature and preterm birth in Detroit. Although the research topic is not new, the authors incorporated the novelty of disentangling the direct from the total effects using DAGs. The study can be considered relevant to clarify the role of ambient temperature as trigger of preterm birth, in particular, to assess to which extent the observed association is due to a mediating effect of air pollutants.

The manuscript is well written and well structured, and the references are appropriate. Given that I don't consider myself an expert on DAG, I will focus my review on the time-series modelling approach and the discussion of the results.

I agree with the authors that time series analysis is an appropriate study design to address their research question. Overall, I consider that the analysis that the authors performed is correct. However, I wonder why the authors did not consider including a denominator on the pregnancies-at risk

(see in Darrow et al Epidemiology 20, 689–698. 2009, Schifano et al Env Int 2014, Vicedo-Cabrera et al Env Res 2014). It has been shown that the daily pool of individuals at risk (ongoing pregnancies at risk of being preterm) changes from one day to the other. By don't taking this into account, one might under or overestimate the association. I would suggest the authors to consider running a sensitivity analysis incorporating as denominator the number of daily pregnancies at risk (or even consider the "corrected" denominator developed in Vicedo-Cabrera 2015).

This is a good point and an important limitation. In the final data set made available to us, we did not have sufficient information to calculate the "pregnancies at risk" population or even a corrected denominator. This calculation depends on knowing the gestational ages of each pregnant woman, and the final data set provided to us included only whether the birth was pre-term or not pre-term. Therefore, we varied the degrees of freedom in the seasonal term to attempt to model changes in the at-risk population as a function of seasonal changes in PTB, as is commonly done in studies of temperature and mortality where daily changes in the at-risk population are also unobtainable.

We modified Methods text to clarify this:

"In the final data set made available for analysis, births were categorized as term vs. PTB, defined as births that were less than 37 gestational weeks. We additionally limited the data set to births that occurred May-September, in order to focus on heat exposures."

In the Discussion, we have added text discussing this limitation:

"Limitations include the fact that we were not able to distinguish between spontaneous and medically indicated PTBs. Considering that medically indicated preterm deliveries are unlikely to be related to temperature, this limitation affects the generalizability of our relative risks to populations where the relative percentages of spontaneous vs. medically indicated PTB differ. We also did not have information on how much earlier the birth was, e.g., a one-day premature birth vs. a six-week premature birth. This prevented us from including an offset for the population of pregnancies at-risk of PTB {Vicedo-Cabrera, 2014 #2420}. Our model included a spline for year and spline for day-of-year to attempt to capture within-season variations in PTBs. Varying the degrees of freedom in the day-of-year term from 2 to 8 had only a mild effect on the relative risks, although this would not have captured sub-weekly changes in PTBs."

The authors considered lag 01 of AT as main exposure term. It has been shown that the association can last up to 2-5 days after the exposure. I would suggest the authors to explore additional lagged terms as sensitivity analysis.

Thank you for this suggestion. We do not include lag 2-5 effects because we would expect potentially protective effects at these lag days given the fact that the pool of women susceptible to preterm birth would be reduced following a day of high heat and, as we explain above, we could not model these daily fluctuations in the at-risk population. We would therefore not know whether a null cumulative effect over lag days 0-5 was due to an acute reduction in the pool of individuals at risk artificially reducing the risk of PTB because we were not reducing the denominator

(referred to as mortality displacement in the temperature-mortality literature) or a true protective effect in the days following high heat, perhaps as women took action to protect themselves in these subsequent days.

To the Discussion, we added:

Varying the degrees of freedom in the day-of-year term from 2 to 8 had only a mild effect on the relative risks, although this would not have captured sub-weekly changes in PTBs.

Minor comments:

- Why the authors considered apparent temperature as temperature index, and not mean temperature or maximum apparent temperature, for example? This should be clarified in the text.

We have added to the Methods:

*“To better represent thermal discomfort, we used apparent temperature (AT) rather than air temperature. AT is similar to heat index, and increases with both air temperature and relative humidity. AT was calculated using the following formula: $[-2.653 + (0.994 * \text{Temperature in } ^\circ\text{C}) + 0.0153 * (\text{Dew Point Temperature in } ^\circ\text{C})^2]$ {Kalkstein, 1986}.”*

- Figure 1 is a bit confusing; I would suggest changing the layout, possibly including colours.

We have modified Figure 1, changing the layout and including abbreviations for our hypothesized confounders and mediators as well as dashed lines of varying sizes to indicate the hypothesized direct and indirect pathways.

- It would be interesting to show the exposure-response curve representing the association.

We have included a plot, as Figure 2, in which we modeled mean AT as a natural cubic spline with 3 degrees of freedom. To the Results, we have added:

“In a crude model of the association between PTB and AT where AT was modeled flexibly as a b-spline with 3 df, we found a nonlinear association, with approximately null effects below 18-19 C and an increasingly stronger positive association at higher temperatures (Figure 2).”

- The study period used is relatively old. One can expect that the observed impacts might have increased due to the increasingly warmer temperatures due to climate change. I would suggest the authors to discuss this point as an additional limitation.

We are actually presenting risks and attributable fractions for a single day of temperature, 24.9 C, and not for all the days that met or exceeded that threshold. Therefore, warmer temperatures over time would not affect the results we presented. To clarify this, we have replaced “24.9 C days” and “18.6 C days” with “a 24.9 C day” and “an 18.6 C day,” respectively, in the Table 3 and 4 titles and in the Results.

- Include a new reference for your interest: Sun et al. Ambient temperature and preterm birth: A retrospective study of 32 million US singleton births. Env Int 2019.

We have added these findings in our list of positive associations in the Discussion text:

“Regions in which these associations have been found include Central Australia (8.3% at 40 °C daily maximum temperature){Mathew, 2017 #2304}, Northern California, USA (11.6% increase for a 5.6 °C increase in weekly average AT in the warm season){Avalos, 2017 #2266}, Alabama, USA (32.4% increase with two consecutive days of daily mean temperatures above the 98th percentile){Kent, 2013 #1279}, Barcelona, Spain (5-day reduction in average gestational age with heat index above the 99th percentile){Dadvand, 2011 #1699}, Rome, Italy (1.9% increase per 1 °C increase in maximum AT in the prior two days and 19% increase during heat waves){Schifano, 2013 #1266}, Brisbane, Australia (13%-100% increase during heat waves), in an aggregated sample of 12 U.S. cities (12-16% increase with 2.8 °C increase in prior week){Ha, 2017 #2234}, in Southern China (7% increase with

previous-week temperatures above 95th percentile){Guo, 2018 #2378}, and in a multi-city USA sample (2% increased PTB risk with extreme heat in the prior 4 days) {Sun, 2019 #3122}.”

Reviewer #3:

This study has many strengths, starting with its research question, which is to assess whether higher temperatures affect risks of preterm birth. Given climate change and the importance of preterm birth as a public health issue, this question merits thorough investigation. The authors add to the previous literature by exploring whether air pollution is a mediator of the relationship between air temperature and preterm birth risk. Their data spanned 10 years between 1991 and 2001 in the Detroit area. This is a period when air conditioning was not as widely spread and this is also an interesting feature of this study.

The manuscript is well written and the authors do a good job of putting their results into context using the scientific literature on temperature and preterm birth risk in the discussion. I am not a specialist in environmental epidemiology or the use of time series methods, so I cannot provide a review of these methods.

As someone who is not a specialist in the methods used in the paper, I would have appreciated more time given to presenting the models, with further explanation of the methodological choices adopted (one ex. “generating three imputations following two burn-ins.” - Why 3 and 2?).

We agree that this deserves more explanation. We have modified the Methods as:

“To account for missing air pollutant values, we conducted multiple imputations using chained equations. We used a more general model including lag days 0-2 of the above meteorological and pollutant values. The air pollutant values were well-predicted by lag days 0-2 meteorology and air pollution values, and from examinations of trace plots, or scatter plots of successive parameter estimates, we determined that a total of three imputations following two burn-ins was sufficient. All subsequent analyses were conducted on each of the three resulting data sets.”

In particular, more explanation about the analyses of seasonal effects would be helpful. I did not understand whether the approach takes into consideration the seasonality of preterm birth or the seasonality of births more generally, or both. I would have liked more explanation of the test for this: (i.e. varying the df in the day-of-year term, using 2, 5, and 8 df).

Thank you for this suggestion. The day-of-year term is meant to capture seasonal variation in PTB counts, which are function of both PTB risk seasonality as well as birth seasonality. As you probably suspect, and as reviewer #3 pointed out, it is a limitation to lump these two concepts into a single term. As we have explained to reviewer #3, we cannot overcome this limitation with the data set that we presently have, because we lack a good way to estimate the “pregnancies at risk” denominator as some birth data sets have. To the Methods, we have changed “effects” to “variations in PTB counts”:

“Therefore, we used a time series design with Poisson regression, controlling for seasonal and long-term variations in PTB counts with a cubic b-spline for day-of-year, with 5, 2 or 8 knots, and a cubic b-spline for year with 2 knots.”

To the Discussion, we added:

“We also did not have information on how much earlier the birth was, e.g., a one-day premature birth vs. a six-week premature birth. This prevented us from including an offset for the population of pregnancies at-risk of PTB {Vicedo-Cabrera, 2014}. Our model included a spline for year and spline for day-of-year to attempt to capture within-season variations in PTBs. Varying the degrees of freedom in the day-of-year term from 2 to 8 had only a mild effect on the relative risks, although this would not have captured sub-weekly changes in PTBs.”

I also wondered how the highest temperatures were distributed across the months included in the study.

We added monthly and yearly time period statistics to Table 1. To the Results, we added:

“There were 9,053 singleton PTBs in this Detroit-area sample, May-September, 1991-2001 (Table 1). There were fewer PTBs in September compared to the other months, and consistent with Detroit’s population decline, the number of PTBs declined with time. Considering individual characteristics, 27.0% of the PTBs were among white mothers and 71.1% were among black mothers. A majority of the mothers were non-smokers (73.1%), had prenatal care (55.6%), were 20-29 years of age (53.2%) and had at least a high school education (60.2%) (Table 1).”

I would also suggest that as the case-crossover design is often used for these analyses and the authors did carry out this analysis, their results should be included in a supplementary table. Also, having these estimates available would be good as a point of comparison with previous studies.

Thank you for this suggestion. The case-crossover results have been added as Supplementary Table S1.

Given that air pollution was not a mediator for the impact of air temperature, did the authors find that air pollution had an impact on PTB risk in this population independently of AT?

We did not examine short-term associations between air pollution and PTB in this paper beyond the indirect effects analysis, but previous research using this data set found effects of some of the pollutants in the third trimester, which the lag 0-1 exposures would fall into. To the Methods, we have added:

“In this DAG, primary and secondary air pollutants are the hypothesized mediators of the temperature-PTB association given that: 1) they have been associated with the health outcome in other locations{Sun, 2015} and in this dataset in the third trimester{Le, 2012}, 2) high temperature enhances the rate of ground-level ozone formation, and 3) power plant emissions increase when temperatures rise to accommodate additional demand for air conditioning{Abel, 2017}.”

Unfortunately the authors are not able to separate medically indicated from spontaneous preterm deliveries (probably about 20-30% of the total) which are less likely to be impacted by temperature (given the hypothesized mechanisms); This is mentioned in the limitations, but could the authors go further and indicate how they believe this limitation affects their estimation of the effect size and attributable risk?

This limitation affects the generalizability of the RRs. For example, in a population where the total number of spontaneous preterm deliveries is higher, the RR would be higher. To the Methods, we have added:

“Limitations include the fact that we were not able to distinguish between spontaneous and medically indicated PTBs. Considering that medically indicated preterm deliveries are unlikely to be related to temperature, this limitation affects the generalizability of our relative risks to populations where the relative percentages of spontaneous vs. medically indicated PTB differ.”

Finally, given the error of gestational age estimation using LMP in US birth certificate data, shown to be larger among Black women, how do the authors assess that misclassification error might affect their results?

LMP tends to result in more births being classified as preterm, particularly in Black women. This exposure misclassification would potentially bias our results towards the null because it is non-differential: on both high and normal AT days, we would have an overestimation of the number of PTBs. For example, if a hot day had 4 PTBs and a normal AT day had 3 PTBs, but the true number of PTBs on each day was actually 3 and 2, respectively, then the estimated RR of 1.33 is lower than the true RR of 1.5. To the Discussion, we have added:

“Another limitation is our dependence on last menstrual period rather than ultrasound-derived gestational age. Last menstrual period tends to result in more births being classified as preterm, particularly among Blacks {Wingate, 2007 #3230}. This limitation, assuming it was not correlated with temperature, would bias our results towards the null.”

Minor comments

Would it be possible to have the distribution by month in Table 1 and some information on trends over the 10 year period in PTB and temperature?

As you suggest, these have been added.

Do the authors mean “all” instead of “any.” “which blocked any of the paths in the DAG from AT to PTB that did not pass through the mediators (Figure 1)” and is this sentence needed? –it seem to be referring to consideration of classical confounding?

We have changed “any” to “all.” Yes, this is referring to classical confounding. See our next response for clarification of the DAG and modeling approach.

One suggestion might be to indicate the variables in the DAG which the authors consider in their model and those which they don't.

We have revised Figure 1 and our description of the DAG as follows:

“A directed acyclic graph (DAG) was constructed to define the causal framework and elucidate potential mediating effects of air pollutants in the association between AT and PTB (Figure 1). In this DAG, AT is the exposure of interest and PTB the outcome of interest. Given suspected associations between PTB and precipitation {Yu, 2018} and between PTB and vitamin D, for which one source is solar radiation {Dovnik, 2018}, as well as causal meteorological relationships between AT, solar radiation, and precipitation via a parent weather system, precipitation and solar radiation are potential confounders of the AT-PTB association. In this DAG, primary and secondary air pollutants are the hypothesized mediators of the temperature-PTB association given that: 1) they have been associated with the health outcome{Sun, 2015}, 2) high temperature enhances the rate of ground-level ozone formation, and 3) power plant emissions increase when temperatures rise to accommodate additional demand for air conditioning{Abel, 2017}. Furthermore, the weather system is a causal parent to both AT and the pollutants and therefore a potential confounder given that meteorological conditions such as wind speed, precipitation, and cloud cover affect pollutant formation and exposure as well as sunlight, which increases AT and promotes ozone formation{Environmental Protection Agency (EPA), 2018}. Therefore, a model characterizing the total effect of AT would need to include the confounders of solar radiation, wind speed, and precipitation but not include pollutants. A model characterizing the direct effect of AT, independent of air pollution mediation, would need to include these mediators. The modeling used to characterize direct and indirect effects is further described in the Statistical Analysis section below.”

Note: I did not see ethics approval mentioned in this document. The data were publically available (as stated in the data sharing agreement section), but does the data sharing agreement require IRB approval. Please clarify given patient location identifiers.

To the Patient and Public Involvement section, we added:

“ . . . This research was deemed exempt from review by the University of Michigan Institutional Review Board.”

We modified the Data Sharing Statement as:

“ . . .These confidential data may be obtained from the MDHHS Division for Vital Records and Health Statistics following completion of a data use agreement and IRB approval. . . .”

Reviewer #4:

The paper examines the effects of the prior two-day mean apparent temperature (AT) on PTB and interaction between AT and maternal characteristics on the risk of PTB. The paper was written in a way that is difficult to follow and provide specific comments. There seems to be a lot of major methodological issues. Below are some examples.

1. The major weakness is the lack of measurement of individual exposure to high temperature, and

control for other factors that affect indoor temperature to which the women were directly exposed, even the outdoor temperature was the same.

As we stated in the Strengths and Limitations to This Study section, although we have individual-level daily exposure, it is actually outdoor citywide exposure rather than indoor exposure. We are not familiar with any preterm birth and temperature studies that actually have indoor temperature exposures, so this is a weakness in the field in general. However, outdoor and indoor temperatures are moderately to strongly correlated (e.g., Nguyen 2014), so a finding of a positive association between outdoor temperature and any health effect, though potentially underestimated, is plausible.

Nguyen JL, Schwartz J, Dockery DW. 2014. The relationship between indoor and outdoor temperature, apparent temperature, relative humidity, and absolute humidity. Indoor Air 24:103-112.

2. Causal framework (Figure 1) is too complex and difficult to interpret. It is unclear how this framework was operationalized in the analysis. For example, what are potential effect modifiers, moderators, or mediators for the association between AT and PTB.

We agree that this figure is confusing, and we have modified it, changing the layout and including abbreviations for our hypothesized confounders and mediators as well as dashed lines of varying sizes to indicate the hypothesized direct and indirect pathways. In response to these reviewers, the following modifications were made to explain the DAG further.

“A directed acyclic graph (DAG) was constructed to define the causal framework and elucidate potential mediating effects of air pollutants in the association between AT and PTB (Figure 1). In this DAG, AT is the exposure of interest and PTB the outcome of interest. Given suspected associations between PTB and precipitation {Yu, 2018} and between PTB and vitamin D, for which one source is solar radiation {Dovnik, 2018}, as well as causal meteorological relationships between AT, solar radiation, and precipitation via a parent weather system, precipitation and solar radiation are potential confounders of the AT-PTB association. In this DAG, primary and secondary air pollutants are the hypothesized mediators of the temperature-PTB association given that: 1) they have been associated with the health outcome{Sun, 2015}, 2) high temperature enhances the rate of ground-level ozone formation, and 3) power plant emissions increase when temperatures rise to accommodate additional demand for air conditioning{Abel, 2017}. Furthermore, the weather system is a causal parent to both AT and the pollutants and therefore a potential confounder given that meteorological conditions such as wind speed, precipitation, and cloud cover affect pollutant formation and exposure as well as sunlight, which increases AT and promotes ozone formation{Environmental Protection Agency (EPA), 2018}. Therefore, a model characterizing the total effect of AT would need to include the confounders of solar radiation, wind speed, and precipitation but not include pollutants. A model characterizing the direct effect of AT, independent of air pollution mediation, would need to include these mediators. The modeling used to characterize direct and indirect effects is further described in the Statistical Analysis section below.”

3. Method section mentioned a number of Confounders and Mediators (page 6), but what variables that were actually treated as Confounders and Mediators in the statistical analysis were not described. In Table 3, only covariates were listed, without defining what variables were confounders and what variables were mediators. No potential effect modifiers were described in the methods. *To clarify which variables were confounders and mediators, we have amended our DAG and its description as described above. To the Methods, we have made the following modifications to draw attention to our explanations of potential effect modifiers and analytic approach:*

“Birth certificate data also included date of birth and maternal age group (16-19, 20-29, and over 30), race (black or white), smoking status (smoker vs. non-smoker), education level (less than high school vs. high school or higher), and level of prenatal care (prenatal care vs. late or no prenatal care), which were used in analyses of effect modification.”

“To understand how the effects varied among maternal subgroups, or effect modification of the PTB-AT association, we simultaneously included interaction terms between the AT spline and the indicator variables for black race, age 16-19, age 30 years or older, low prenatal care (late or no prenatal care), current smoker, and no high school education. We expanded the data set such that we had rows for each unique combination of date and daily exposures, race, age group, education, prenatal care, and

smoking status. Given the large number of zero counts, we specified a negative binomial distribution rather than a Poisson distribution. For public health significance, we were interested in the absolute rather than the relative increase in PTB risk due to synergistic effects between temperature and each modifier of interest. Therefore we focused on additive, rather than the multiplicative, interactions. Relative excess risk due to interaction (RERI){Nie, 2010 #2436} was calculated for each potential modifier as:

$$RR_{AT=95th,EM=1} - RR_{AT=95th,EM=0} - RR_{AT=50th,EM=1} + 1 \quad \text{Equation 2}$$

where EM was the effect modifier of interest and the denominator of each RR was the risk at the 50th percentile of AT (18.6 °C) and the absence of the modifier (EM = 0). RERI confidence CIs were bootstrapped from 1,000 samples.”

4. Exposure definition: ‘prior two-day’ mean apparent temperature (AT) was mentioned in the Abstract, but not in the method. Furthermore, when the temperature was measured and why it would be related to pregnancy and the occurrence of PTB was not stated. Was the temperature measured when the women was pregnant? If so, during what period of pregnancy?

Thank you for this comment. To clarify that we used as our exposure the temperatures on the day of and the day before birth, we have added to the Methods:

“Therefore, we used a time series design with Poisson regression, controlling for seasonal and long-term variations in PTB[CG1] counts with a cubic b-spline for day-of-year, with 5, 2 or 8 knots, and a cubic b-spline for year with 2 knots. AT exposure was characterized as two-day mean AT, or the mean of the AT values on the day of and the day before birth. “

5. It seems that the paper tested additive (as opposed to multiplicative) interaction between AT and maternal characteristics on the risk of PTB. This should be clearly described in the method.

To clarify that this was the reason we used RERIs, we have made the words “additive” and “multiplicative” italicized in the Methods.

VERSION 2 – REVIEW

REVIEWER	Qihong Deng Central South University, China
REVIEW RETURNED	16-Nov-2019
GENERAL COMMENTS	The authors addressed my concerns and I am satisfied with the author's responses. Please provide the literature in the references for "High ambient temperatures in the month of conception and the third trimester were also positively associated with PTB in Changsha, China {Zheng, 2018 #3222;Zhong, 2018 #3223}".
REVIEWER	Ana M Vicedo-Cabrera Institute of Social and Preventive Medicine
REVIEW RETURNED	20-Nov-2019
GENERAL COMMENTS	The authors have addressed my suggestions and comments appropriately.
REVIEWER	Duc Anh Ngo Nashville Metro Public Health Department USA
REVIEW RETURNED	22-Nov-2019

GENERAL COMMENTS	The authors have addressed comments.
--------------------------------------

VERSION 2 – AUTHOR RESPONSE

Reviewer 1

Please provide the literature in the references for "High ambient temperatures in the month of conception and the third trimester were also positively associated with PTB in Changsha, China {Zheng, 2018 #3222;Zhong, 2018 #3223}".

Corrected.